# Phylogenetic Diversity of Wetland Plants across China

**DOI:** 10.3390/plants10091850

**Published:** 2021-09-06

**Authors:** Aiying Zhang, Zhixia Ying, Xunyu Hu, Mingjian Yu

**Affiliations:** 1College of Life Sciences, China Jiliang University, Hangzhou 310018, China; ayzhang@foxmail.com; 2MOE Key Laboratory of Biosystems Homeostasis & Protection, College of Life Sciences, Zhejiang University, Hangzhou 310058, China; 3School of Life Science, Nanchang University, Nanchang 330031, China; yingzhixia@126.com; 4East China Inventory and Planning Institute, National Forestry and Grassland Administration, Hangzhou 310019, China; huxunyucooky@163.com

**Keywords:** wetland restoration, invasive, dominant, clustered, species screen

## Abstract

Accelerating and severe wetland loss has made wetland restoration increasingly important. Current wetland restorations do not take into consideration the ecological adaptability of wetland plants at large scales, which likely affects their long-term restoration success. We explored the ecological adaptability, including plant life forms and phylogenetic diversity, of plants across 28 wetlands in China. We found that perennial herbs were more common than annual herbs, with the proportion of perennial herbs accounting for 40–50%, 45–65%, 45–70%, 50–60%, and 60–80% of species in coastal wetlands, human-made wetlands, lake wetlands, river wetlands, and marsh wetlands, respectively. A ranking of phylogenetic diversity indices (PDIs) showed an order of marsh < river < coastal < lake < human-made, meaning that human-made wetlands had the highest phylogenetic diversity and marsh wetlands had the lowest phylogenetic diversity. The nearest taxon index (NTI) was positive in 23 out of 28 wetlands, indicating that species were phylogenetically clustered in wetland habitats. Dominant species tended to be distantly related to non-dominant species, as were alien invasive species and native species. Our study indicated that annual herbs and perennial herbs were found in different proportions in different types of wetlands and that species were phylogenetically clustered in wetland habitats. To improve wetland restoration, we suggest screening for native annual herbs and perennial herbs in proportions that occur naturally and the consideration of the phylogenetic similarity to dominant native species.

## 1. Introduction

Wetlands, known as the “kidney of the earth”, are one of the world’s three major ecosystems and provide indispensable resources for humans and biodiversity [1]. Wetlands have diverse roles from nutrient cycles (e.g., water, carbon) to human health [2,3,4,5]. However, serious wetland loss has occurred globally, driven by agricultural development, urbanization, fish farming, and climate warming [1,6].

Even without the provision of supporting evidence, Davidson (2014) [7] estimated that 64–71% of global wetlands had been lost since 1900 AD; meanwhile, Hu et al. (2017) [1] showed a 33% decrease in global wetlands until 2009 by calculating the Precipitation Topographic Wetness Index and global remote sensing training samples.

In the past six decades, Flanders (northern Belgium) has lost 75% of its wetland habitats [8], and the wetland area of the Sanjiang Plain—which contains the largest marsh wetland in China—has declined by 73.3% (about 2.77 million ha) since 1954 [9]. Wetland loss severely reduces or eliminates ecosystem services, such as biodiversity loss [8], water availability [4], and greenhouse gas capture [10]. Therefore, we must urgently implement effective wetland restoration projects.

Species screening is one of the central factors affecting the success of wetland restoration. Past wetland restorations have focused on screening for some common aquatic species to improve water quality and landscape values [11,12]. However, most of these efforts have neglected assessing the long-term stability (including the stability of a current situation to resist changes and to recover) of human-made plant communities because ecosystem structure and function recovery is typically slow after restoration [13,14]. Thus, exploring the ecological adaptability of wetland plants at large scales would be beneficial in species screening to improve the probability of successful wetland restoration in the long run.

With an area of 53.6 million ha, wetlands in China account for about 10% of the world’s wetlands [15]. Here, we selected 28 wetlands (including lake, marsh, coastal, river, and human-made wetlands) in China to explore the plant life forms and phylogenetic diversity of wetland plants, with the goal of deepening our fundamental understanding of ecological adaptability of wetland plants.

## 2. Materials and Methods

### 2.1. Study Site and Data Collection

To study the plant life forms and phylogenetic diversity of wetland plant species, we collected species names from published references conducted by field investigations (after year 2000) in 28 wetlands, including 12 lake, 7 marsh, 3 coastal, 3 river, and 3 human-made wetlands (Figure 1). Among these 28 wetlands, 20 are designated as Ramsar Sites (Appendix A).

The Latin names of the species from published references (Appendix A) were corrected according to the Flora of China (FOC, http://foc.iplant.cn/, accessed on 3 July 2021), Catalogue of Life (CoL, http://www.catalogueoflife.org/, accessed on 3 July 2021), The Plant List (TPL, http://www.theplantlist.org/, accessed on 3 July 2021), and Plants of the World Online (POWO, http://www.plantsoftheworldonline.org/, accessed on 3 July 2021). To verify species as wetland plants, we used the China Wetlands Resources Master Volume [15]. We recorded the life forms of species from these published references and FOC. Since there was only a small number of biennials, we classified biennials as perennials to simplify the data analyses. As one of the indicators that determine the physiognomy of plant communities, life form reflects the long-term biological adaptation of plants, which is the result of convergent adaptation to the same environment [16].

### 2.2. Phylogenetic Diversity

With the relation to plant evolutionary history [17], phylogenetic diversity has been applied to explore the mechanism of species co-existence [18]. We built a phylogenetic tree with the package ‘*V.PhyloMaker*’ [19] in R 4.0.3 [20] using a mega-tree of the largest phylogeny for vascular plants found so far. Based on this tree, we calculated two commonly used phylogenetic diversity indices. The phylogenetic diversity index (PDI) measures the total branch lengths of the subtree containing a specific assemblage of taxa [21]. PDI was calculated as PDI = (PD_observed_ − PD_randomized_)/(sdPD_randomized_), where PD_observed_ was the observed PD, and PD_randomized_ and sdPD_randomized_ were the expected mean and standard deviation of the species of the randomized assemblages. The nearest taxon index (NTI) is the standardized effect size of the mean phylogenetic distance to the nearest taxon for each taxon (MNTD) in the assemblage and thus estimates the extent of terminal clustering, independent of deep-level clustering [22]. NTI was calculated as NTI = − (MNTD_observed_ − MNTD_randomized_)/(sdMNTD_randomized_), where MNTD_observed_ was the observed MNTD, and MNTD_randomized_ and sdMNTD_randomized_ were the expected mean and standard deviation of all nearest pairs of the randomized assemblages. PDI and NTI were both calculated under null models that maintained species richness, conducted by the functions *pd.query* and *mpd.query* in the R package ‘*PhyloMeasures*’ [23], respectively. Specifically, we put all species at the 28 wetlands into a species pool. With this pool, we generated 999 simulated null communities for each wetland. In each of the 999 loops of our simulation, we randomly conducted a fixed number of species richness at all wetlands, then calculated the mean and standard deviation of the phylogenetic diversity indices of randomized assemblages. Larger PDI values indicate a higher phylogenetic diversity. A negative NTI value indicates that the observed MNTD is greater than the value of MNTD in randomized assemblages—i.e., species are more distantly related than randomized assemblages, and thus species are phylogenetically over-dispersed. In contrast, a positive NTI value means that observed MNTD is less than the value of MNTD in randomized assemblages—i.e., species are more closely related than randomized assemblages, and thus species are phylogenetically clustered.

The phylogenetic distances between wetlands were visualized by Principal Co-ordinates Analysis (PCoA) with the R package ‘*FD*’ [24]. The function *comdist* was used to calculate MPD (mean pairwise phylogenetic distance), and the function *comdistnt* was used to calculate MNTD in the R package ‘*picante*’ [25]. The first five PCoA axes for MNTD explained 90% of the variance. However, the detected patterns of MPD were mostly random [26], since the first two axes explained less than 10% of the variance and the first five PCoA axes explained 23% of the variance. Thus, we did not use MPD for quantifying phylogenetic diversity.

We compared PDI and NTI among different wetland types (lake, marsh, coastal, river, and human-made), since common species were significant and always preferentially selected for wetland restoration practices. We also compared the PDI and NTI between common species and non-common species in all wetlands. Meanwhile, considering that a considerable number of aquatic plants have been introduced into China [27], we compared the PDI and NTI between invasive species and non-invasive species.

## 3. Results

We found 2204 vascular plant species (belonging to 795 genera in 169 families) in the 28 studied wetlands. Perennial herbs and annual herbs co-dominated the wetland habitats (Figure 2), with perennial herbs being more frequent. The proportions of perennial herbs accounted for 40–50%, 45–65%, 45–70%, 50–60%, and 60–80% of species in coastal, human-made, lake, river, and marsh wetlands, respectively (Figure 2). Marsh wetlands had the highest proportion of perennial herbs, while coastal wetlands had the lowest.

The species in Asteraceae, Poaceae, and Cyperaceae families were co-dominant in the study wetlands, accounting for 30% of all species (Appendix A). There were 250 dominant species (belonging to 164 genera in 66 families); among them, 40% of the species were in Cyperaceae, Poaceae, and Asteraceae (Appendix A). There were 102 alien invasive species (belonging to 77 genera in 27 families), and half of them were in Asteraceae, Poaceae, and Amaranthaceae (Appendix A). Annual herbs and perennial herbs accounted for 18% and 65% of the dominant species, respectively (Appendix A). For alien invasive species, annual herbs and perennial herbs accounted for 60% and 31%, respectively (Appendix A).

The species richness of different wetland types was ranked as river < marsh < coastal < human-made < lake (Figure 3a), while PDI was ranked as marsh < river < coastal < lake < human-made (Figure 3b). Thus, human-made wetlands had the highest phylogenetic diversity and marsh wetlands had the lowest phylogenetic diversity. For 23 out of 28 wetlands, the NTI values were positive, indicating that species were phylogenetically clustered in wetland habitats (Figure 3c). PDI was strongly negatively correlated with NTI (Pearson correlation coefficient = −0.967, *p* < 0.001). However, species richness was not significantly correlated with PDI (Pearson correlation coefficient = 0.338, *p* = 0.008), nor NTI (Pearson correlation coefficient = −0.308, *p* = 0.111).

In the 28 wetlands, the NTI value of dominant species was positive—i.e., the dominant species were phylogenetically clustered, as were the alien invasive species (Appendix A). The MNTD value between dominant species and non-dominant species was large (Appendix A), indicating that dominant species were phylogenetically distant from non-dominant species. The same was found for the MNTD value between alien invasive species and non-invasive species (i.e., native species; see Appendix A).

## 4. Discussion

### 4.1. Perennial Herbs

The higher frequency of perennial herbs in wetland habitats is consistent with previous findings, such as in agricultural landscapes after wetland restoration in the USA [28] and 74 different kinds of wetlands in Canada [29]. Perennials, such as *Carex* spp. and *Phragmites australis*, tend to have larger root systems that may augment the ecosystem’s adaptability to water level fluctuations and tolerance to long-term seasonal submergence. Many perennials can spread or propagate vegetatively, which may also benefit wetlands with fluctuating conditions. Annuals, with smaller root systems and typically without vegetative reproduction, may be at a disadvantage in wetland habitats.

### 4.2. Phylogenetic Diversity

It was not surprising that wetland species were phylogenetically clustered, as Asteraceae, Poaceae, and Cyperaceae are among the largest plant families (Appendix A). However, it was interesting that dominant species tended to be phylogenetically distinct from non-dominant species, which might impact wetland restoration strategies.

In China, marsh wetlands had a low species richness and phylogenetic diversity relative to other wetland types. This may be because most marsh wetlands were distributed in the northeast region, Yunnan–Guizhou region, and Qinghai–Tibet Plateau, which have colder climates and lower diversity [30]. It is also possible that the diversity was low in marsh wetlands because they occupy only small areas, usually distributed near rivers and lakes [30]. Consistently, marsh wetlands had a lower phylogenetic diversity compared to the Gobi, salt marshes, and sand dunes in the Jiayuguan Caohu Wetland of China [31].

Lake and human-made wetlands had a higher species richness and phylogenetic diversity. One explanation for this might be that these two kinds of wetlands were mainly distributed in the southeast part of China, which has a warmer climate and a higher regional species richness [32]. Moreover, human activities are extremely frequent in the southeast part of China, which likely leads to more aquatic species being planted and alien species being introduced, such as *Eichhornia crassipes* (water hyacinth) and *Alternanthera philoxeroides* (alligator weed).

Many tree species have been planted at the junctions between wetlands and adjacent landscapes, which might contribute to the high diversity of human-made wetlands. However, some tree species can lead to wetland degradation. For example, rapid-growth species of *Populus* were introduced into Dongting Lake, the second largest freshwater lake in China, for forestry and paper [33]. *Populus*, also known as “the pump of the wetland” [33], accelerated the drying of the wetlands and changed the structure of the wetland soil. To protect the Dongting Lake wetland, the local government implemented the removal of *Populus* plants in 2017 [33]. To avoid the high cost of tree removal projects, as seen at Dongting Lake, it is important to follow ecologically sound vegetation addition practices.

Here, alien invasive species were phylogenetically distinct from native species in wetland habitats (Appendix A). Consistently this is supported by a functional trait approach, where the differences between alien invasive species and native species improve the success of invasiveness [34]. A total of 478 aquatic plants species have been introduced to China, 95% of which were introduced for their landscape value, such as ornamental and soil and water conservation [27]. For instance, *Eichhornia crassipes* (water hyacinth) and *Alternanthera philoxeroides* (alligator weed) have already led to the destruction of ecological services and caused severe economic losses [35]. However, alien mangrove plants, such as *Sonneratia apetala* and *Laguncularia racemosa*, have restored ecosystem services in mangrove forests without causing invasions in China [36]. We must be very cautious when introducing species and fastidiously monitor the impact of introduced species on local ecosystems.

### 4.3. Suggestions on Wetland Restorations

By exploring the life forms and phylogenetic diversity of wetland plants, we offer some suggestions for species screening in wetland restoration to maximize the ecological, landscape, and environmental values of wetlands. We suggest to (1) screen for native annual herbs and perennial herbs in proportions that occur naturally; (2) consider the phylogenetic similarity to dominant native species: for instance, we could screen for plant species that are phylogenetically similar to local dominant species to improve the probability of successful wetland restoration in the long run.

## 5. Conclusions

In 28 wetlands of China, we found that annual herbs and perennial herbs were distributed in different proportions in different types of wetlands, and that wetland plant species were phylogenetically clustered. Our study contributes to the fundamental understanding of the ecological adaptability of wetland plants, which should be considered during wetland restoration. For further studies, exploring the functional and phylogenetic diversity of wetland species and adjacent landscape species would offer deeper a understanding of the ecological adaptability of wetland plants.

## Figures and Tables

**Figure 1 plants-10-01850-f001:**
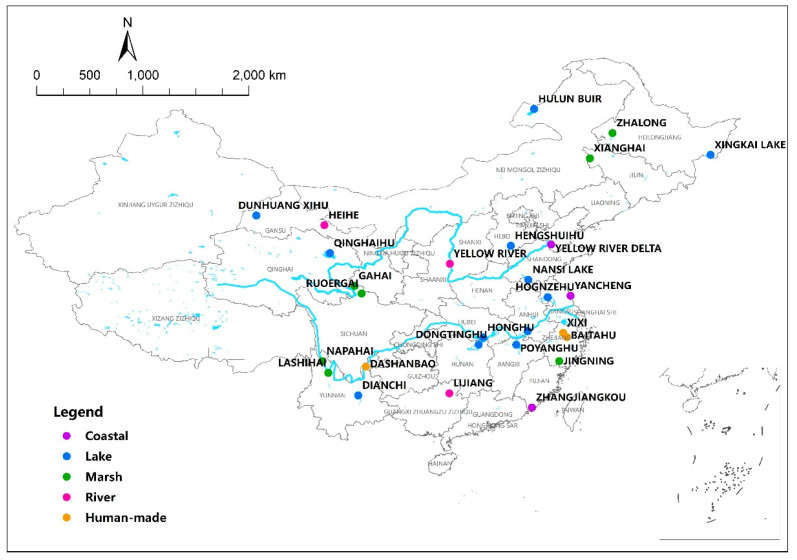
Locations of the 28 wetlands in this study in China.

**Figure 2 plants-10-01850-f002:**
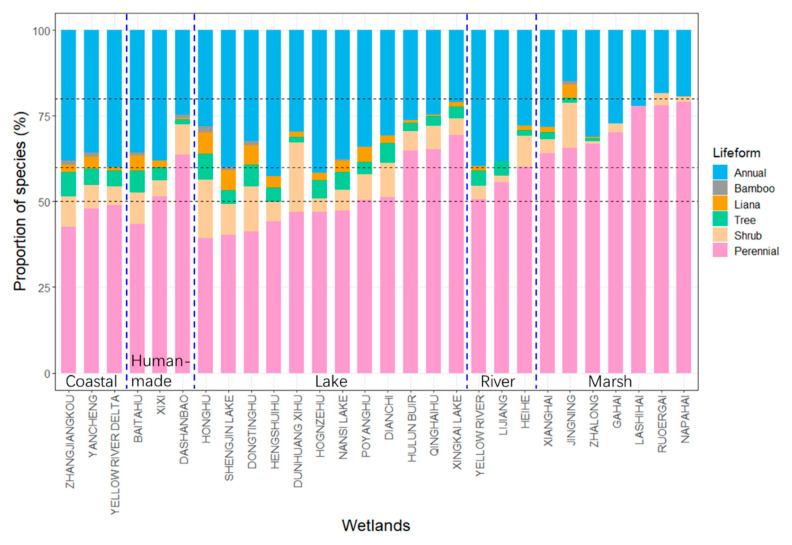
Proportions of wetland plant species of different life forms in the 28 wetlands studied.

**Figure 3 plants-10-01850-f003:**
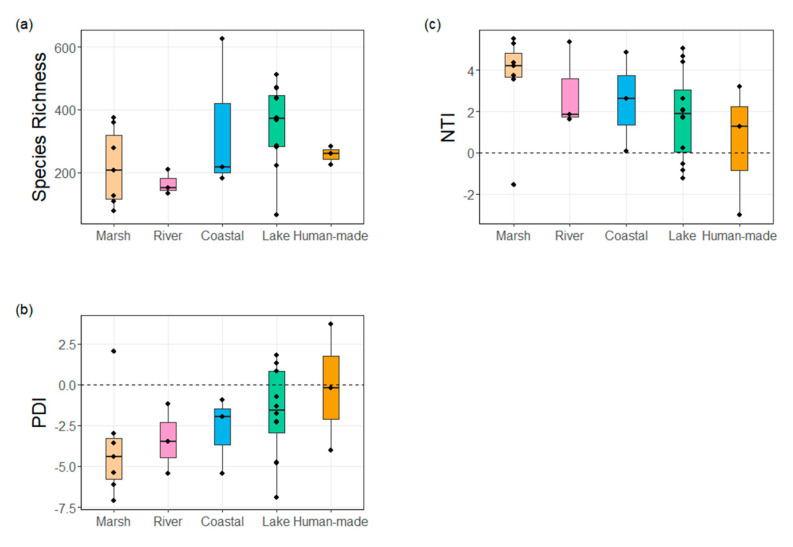
Species richness and phylogenetic diversity indices of wetland types.

## Data Availability

The data are provided in the Appendix A.

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
