# Peer review of "Phylogenetic Diversity of Wetland Plants across China"

_plants, 2021, doi:10.3390/plants10091850_

Round 1

Reviewer 1 Report

Interesting manuscript but I have found disproportion among the title, abstract and the real content of the manuscript. That is why I cannot recommend it for publishing – see, please, my comments below:

The title should be changed as the manuscript is not strictly about “Phylogenetic Diversity of Wetland Plants” – it is rather about comparison of different characteristics of plants from different types of wetlands. The importance of this manuscript is (according the introduction) in searching for special characteristics of plants that are important for wetlands conservation and restoration.

What is completely missing is conceptual background about importance of researched characteristics (“physiological characteristics and phylogenetic diversity”) for wetland conservation and restoration – please prepare scientific review of this topic after Introduction.

Methods are described well; but I am missing the description of how wetlands were selected – just from simple look at the map it is clear that they come from distinct climatic conditions and this could significantly affect results, especially in the case of human-made wetlands.

Also the description of “physiological characteristics” is missing – this is because no “physiological characteristics” were not discussed further – except trees-perennial-annual. I think that other “physiological characteristics” are needed. Please analyse also other important characteristic for wetland plants – especially differences in physio-ecological characteristics will be important for readers.

Results are described well.

Discussion is poor. The discussion of physiological diversity is extremely poor and not suitable for scientific journal. Comparison of phylogenetic diversity with other biotopes is missing. “Suggestions on Wetland Restorations” must be enlarged.

. . . based on the shortages I have found; my recommendation is “rejection”. But the idea of research is interesting I think that after rewriting it will be suitable for journal Plants.

Reviewer 2 Report

General comments

This is an interesting communication and would be of interest to PLANTS readers. Though a communication is meant to be brief, there are a few places where authors need to add discussion on implications of their results. See below.

Specific comments

Line 60. I don’t think you can claim to have explored the physiological characteristics of wetland plants, which I would consider the functions of plants such as types of photosynthesis, signaling, metabolism. I suggest you use the term “plant life form” as you are categorizing as annual, perennial, tree, etc.

  1. suggest “…20 are designated as Ramsar Sites (Table S1).”
  2. Not sure what you mean by “dated”. Please explain.
  3. change to “…more frequent.”
  4. This sentence does not read well. Suggest: “The same was found for the MNTD value betwee…”
  5. Do you mean multiple Carex species? If so, use Carex spp. If you mean just one unidentified Carex species, use Carex sp.
  6. It was not surprising that…
  7. …diversity relative to other…
  8. which have
  9. Cite a reference that more northern, colder climates have lower diversity.
  10. ...human activities are extremely…
  11. be consistent and add a common name for Alternathera philoxeroides (alligator weed)
  12. be explicit about why Populus were introduced to Dongting…was it for forestry/paper/shade?
  13. citation needed for “pump of the wetland”.
  14. Not sure what you mean by natural pattern of science. Do you mean “To avoid the high cost of tree removal projects like seen at Dongting Lake, it is important to follow ecologically sound vegetation addition practices.”
  15. Not sure what you mean by landscape value. Do you mean they were introduced for landscaping?
  16. we offer
  17. to maximize
  18. We thank for….We would like to thank Ian…

References are numerical at this point, but listed by author in text.

Reviewer 3 Report

The authors present a study of plant phylogenetic diversity of wetlands in China based on physiological characteristics and phylogenetic indices. Through this analysis they intend to understand the ecological adaptability of wetland plants to improve restoration efforts. However, I would insist authors to make this argument and the relationship between physiology-phylogenetic diversity-ecological adaptability clearer throughout the text because in my opinion a detailed explanation lacks. Reporting also for phylogenetic distance of dominant vs non-dominant or alien vs native and their relationship with ecological adaptability seems vague in the ms. It needs more explanation in my opinion. The ms is well written in an understandable English. The objectives of the study are quite clear, and the data set is appropriate to study these objectives. Nevertheless, I think that there are a couple of methodological shortcomings that I should report. In addition, it seems to me that the cited references are updated.

Comments

Line 53: again, the term “long-term stability” needs more details

Line 83: “Based on this tree”

Methods: to account for the effect of species richness on phylogenetic indices the authors calculate standardized effect sizes (PDI, NTI). However, the authors do not explicitly describe how the null models have been calculated. The authors report that the null communities maintain species richness. Did the authors check how “building” null communities based on other ways (e.g. considering commonness and rarity of each species or sampling species from communities from different wetland types)?

Line 98: than “the value of MNTD in randomized assemblages”

Line 112: We found 2204…

Line 160: …which might impact…. Why? The authors could add a sentence with more details

Round 2

Reviewer 1 Report

I am very sorry that I did not realized that it is not Article in my first review. As Communication it cannot be extended with other physiological characteristics.

My other comments have been taken into account by authors in the new version, so I can recommend the acceptance of the manuscript.